# Physiological Responses to Combat Sports in Metabolic Diseases: A Systematic Review

**DOI:** 10.3390/jcm11041070

**Published:** 2022-02-18

**Authors:** Max Lennart Eckstein, Melanie Schwarzinger, Sandra Haupt, Nadine Bianca Wachsmuth, Rebecca Tanja Zimmer, Harald Sourij, Paul Zimmermann, Beate Elisabeth Maria Zunner, Felix Aberer, Othmar Moser

**Affiliations:** 1Division of Exercise Physiology and Metabolism, Department of Sport Science, University of Bayreuth, 95447 Bayreuth, Germany; max.eckstein@uni-bayreuth.de (M.L.E.); melanie.schwarzinger@uni-bayreuth.de (M.S.); sandra.haupt@uni-bayreuth.de (S.H.); nadine.wachsmuth@uni-bayreuth.de (N.B.W.); rebecca.zimmer@uni-bayreuth.de (R.T.Z.); paul.zimmermann@uni-bayreuth.de (P.Z.); beate.zunner@uni-bayreuth.de (B.E.M.Z.); felix.aberer@uni-bayreuth.de (F.A.); 2Interdisciplinary Metabolic Medicine Trials Unit, Division of Endocrinology and Diabetology, Department of Internal Medicine, Medical University of Graz, 8036 Graz, Austria; ha.sourij@medunigraz.at

**Keywords:** metabolic disease, type 1 diabetes, type 2 diabetes, metabolic syndrome, overweight, obesity, combat sports, martial arts

## Abstract

The aim of this systematic review was to investigate how individuals with metabolic diseases respond to combat sports and if they are feasible, safe, and applicable. A systematic literature search was conducted in PubMed, from inception until 22 January 2021. Studies were included if combat sport exercise sessions were clearly defined and participants had the following types of metabolic disease: type 1 or 2 diabetes mellitus, metabolic syndrome, overweight, and obesity. Eleven studies, involving 472 participants of all age groups with type 1 diabetes mellitus, metabolic syndrome, overweight, or obesity were included in this systematic review. No studies involving combat sports and individuals with type 2 diabetes were found. Combat sports showed improved HbA_1c_ levels over time in individuals with type 1 diabetes mellitus, which was not significantly different compared to the control group (*p* = 0.57). During the follow-up period, glycaemic variability decreased in those actively participating in combat sports. Fat-mass was higher in athletes performing combat sports with metabolic syndrome, compared to athletes without an increased cardiometabolic risk. In overweight/obese adolescents, combat sports showed improved parameters of physical fitness, cardio autonomic control, strength, and body composition compared to control groups. In all studies included in this systematic review, no adverse event associated with combat sports was reported. In conclusion, combat sports are safe and feasible in individuals with diabetes and/or obesity. For individuals with type 2 diabetes mellitus, no recommendations can be made, due to the lack of evidence in this cohort. Future studies investigating combat sports and metabolic diseases should aim for a structured exercise regimen and acknowledge the experience of the participants prior to starting an exercise intervention involving combat sports.

## 1. Introduction

The prevalence of metabolic diseases, such as the metabolic syndrome (MetS), type 1 diabetes mellitus (T1DM), type 2 diabetes mellitus (T2DM), and obesity has been rising over the past decades [1,2,3]. These diseases have a major impact on the physiology and favor the development of comorbidities that may further deteriorate quality of life, by reducing physical capacity, deteriorating bone quality, and increasing the risk of cardiovascular disease [4,5,6].

For adults aged between 18–64 years, the World Health Organization (WHO) recommends at least 150 min of moderately intense aerobic physical activity or 75 min of vigorously intense aerobic physical activity, or a suitable combination of both every week [7]. Low to moderate physical activity is usually considered any kind of recreational activity of aerobic nature, such as walking, hiking, or cycling.

In addition, the interest in conducting combat sports, defined as physical contact between two opponents in a martial arts manner, has increased rapidly during the last few decades. Many people enjoy doing combat sports for recreational purposes. About 2% of the physically active European population practiced martial arts in 1999, making martial arts the tenth most popular sport next to hill walking/climbing [8]. Moreover, children and adolescents, in particular, practice combat sports, such as taekwondo or judo [9,10,11]. This type of sport is an interval-based physical activity, which is conducted in groups or with a partner. The intensity during the training sessions may vary, yet it is mostly of moderately intense aerobic nature, with short bouts of vigorous anaerobic intensity. At the same time, the positive effects of high-intensity interval training on individuals with metabolic diseases, for improving glycaemia, body composition, and fitness, have already been investigated thoroughly [12,13,14].

The emerging prevalence of metabolic diseases, and the increasing interest of many people in combat sports, lead to the question of whether combat sports, which cover several types of varied intensity training methods, are safe and applicable for people affected by a metabolic disease. Previous studies have shown that combat sports increase physical capacity [15] and improve bone mineral density [16,17].

One might wrongly assume that combat sports have a higher risk of injury such as sprains, fractures, or concussions compared to team sports such as football. However, according to Arriaza et al. and Emery et al., 0.67 injuries occur every 1000 min during combat sports, in comparison to 5.59 injuries every 1000 min during amateur football [18,19]. A further advantage of combat sports compared to other sports is the reduced level of aggressiveness of individuals actively practicing combat sports, due to diffusion of emotions, relieving tension, and improving self-esteem and quality of life [20].

According to Hamasaki et al., no positive effects on metabolic diseases by Tai Chi practice could be proven. In contrast other martial arts performed as Kata, e.g., Kung Fu or Karate, demonstrated some positive effects on body composition, glycaemic control, and arterial stiffness [21]. Due to the specific nature of combat sports (whole body movements at various intermittently conducted intensities) it is of interest if individuals with metabolic disease may benefit from regularly conducting this type of exercise by improving the parameters of metabolic disease and/or reducing the development of comorbidities. Consequently, the aim of this systematic review was to investigate how individuals with metabolic disease respond to combat sports and if they are feasible, safe, and applicable in daily life. Unlike numerous other reviews investigating the interaction of martial arts and metabolic disease with contrary outcomes, the focus of this systematic review will be on combat sports.

## 2. Materials and Methods

This systematic review is reported in accordance with the PRISMA guidelines [22].

### 2.1. Search Strategy and Selection Criteria

We selected relevant studies published from inception until the 22 January 2021 by searching PubMed. We used combined term medical subject headings (MeSH) and text words (tw) to search relevant studies. Search terminology and strategy can be found in the Appendix A. Potentially eligible studies were considered for review if the articles were written in English. Studies were excluded if combat sports were not clearly defined in individuals with metabolic diseases or the disease was not clinically specified. Likewise, we excluded studies if neither the type nor duration of the intervention were defined.

### 2.2. Data Extraction and Quality Assessment

Two independent investigators (M.L.E. and M.S.) reviewed the study titles and abstracts. Studies that satisfied the inclusion criteria were then retrieved for full-text evaluation. The following data were extracted from each selected study: study characteristics (author, publication year, journal country, city continent, type of study), population (metabolic pathology, number of participants, age range, participants completed the study, gender, age of participants that completed the study, anthropometrics), combat sports and performance parameters (exercise type, duration, intensity, frequency). If any of the pre-defined criteria for inclusion in the systematic review were not available, authors were contacted for individual participant data. If the data were not available, studies were excluded from this systematic review.

### 2.3. Data Synthesis and Analysis

A narrative descriptive analysis was performed to summarize the characteristics of studies, such as year of publication, trial design, sample size, country where the study was conducted, age of the participants, types of interventions, and duration of interventions. 

For this review we focused on definition of combat sports according to Krabben et al.: combat sports covers disciplines where two opponents fight in a regulated combat, trying to win over their adversary through usage of strikes, kicks, or grappling in different configurations [23]. The main difference between martial arts and combat sports is the necessarily included physical contact in combat sports, which is not a mandatory requirement in martial arts [24]. The principal items of martial arts are sets of formal exercises and expressive movements, which do not always include physical contact [24]. Consequently for this review, combat sports were defined as any kind of martial arts involving physical contact involving striking, kicking, wrestling, or grappling. Studies including martial arts without physical contact or types of kata against imaginary opponents or stylistic sequences presenting martial arts techniques were excluded. 

Studies were included if the participants either had a clinical diagnosis of type 1 diabetes, type 2 diabetes, metabolic syndrome, overweight, or obesity. If a clinical diagnosis was unspecific and the health status of the participants of the study was not monitored throughout the study, and/or outcome parameters were not health related, the study was excluded from this systematic review.

## 3. Results

After careful evaluation, a total of eleven clinical studies were extracted from 545 articles that met the predefined objectives. The structure of the article selection process is described as a flow diagram in Figure 1. Studies published between 2009 and 2020, enrolling a total of 472 participants, were included. Two studies were conducted in a group of individuals with T1DM [9,25], two studies in individuals with MetS [26,27], and seven studies in individuals who were overweight or obese [11,28,29,30,31,32,33]. No studies were found in which combat sports were investigated in individuals with T2DM.

### 3.1. Type 1 Diabetes Mellitus

The first study was a two-year follow-up of five combat sports athletes with T1DM, who were supervised by clinical experts in an exercise-focused diabetes clinic [9]. The combat sport athletes, two kickboxers and three mixed martial arts (MMA) athletes, were all male and aged between 18 and 34 years. All athletes received a re-education on the general principles of the insulin therapy and each participant received an individual treatment, considering the types of training, the order of exercises, and the ratio of those different exercise types interfering with each other. The collected data were additionally compared with data from individuals with T1DM who exercise less than 1 h daily and do not participate in combat sports. They were matched by sex, age, and glycated hemoglobin (HbA_1c_) levels at baseline. Glycaemic control measured via HbA_1c_ improved and was sustained in all athletes, starting with HbA_1c_ levels of 8.1% and ending with 7.4%. In that study, no standard deviation was provided. However, compared to the control group of individuals with T1DM who did not participate in combat sports, the HbA_1c_ levels of the athletes were not significantly lower at the end of the follow-up period (*p* = 0.57). Second, blood glucose variability decreased from 84 ± 6 to 62 ± 5 mg/dL, but also this result did not reach significance compared to the control group (*p* = 0.378). 

In the second study, three out of the five participants from the previous study from Benebek-Klupa et al. were followed-up for three years, to examine if the glycaemic improvement could be further maintained [25]. Specifically, the HbA_1c_ levels decreased to an average level of 7.1%, without increasing the number of periods with symptomatic hypoglycaemia [25] (Table 1).

### 3.2. Type 2 Diabetes Mellitus

In our literature search, no studies involving combat sports/physical contact sports in T2DM were found. 

### 3.3. Metabolic Syndrome

The study of Guo et al. assessed the prevalence of MetS among Chinese, professional athletes in strength sports, including Judo and Wrestling [26]. Out of 261 athletes (130 females and 131 males) examined in this cross-sectional study, 46 participated in judo, 49 in wrestling, while the other 166 individuals participated in non-combat sports [26]. Participants were divided into two groups: an unlimited body weight group (UWB) and a limited body weight group (LBW), which included all athletes who competed in weight-classes with an upper weight-limit [26]. Anthropometrics (height, body mass, waist circumference, hip circumference), body composition, as well as seated blood pressure, blood glucose, and blood lipids (TG, low density lipoprotein cholesterol, high density lipoprotein cholesterol) were measured [26]. The results demonstrated that the prevalence of the MetS was 31.8% among all athletes, with a significantly higher prevalence in the UBW group [26]. While, 49% of the female athletes and 88% of the male athletes in the UBW group showed characteristics of MetS, no female athlete and only 18% of the male athletes in the LBW group had a MetS [26]. 

Another study aimed to capture the characteristic body composition and cardiometabolic health of Japanese male heavy-weight judo athletes [27]. Overall, 58 athletes were assessed, of which 19 were heavy-weight judo athletes, besides 22 American football and rugby heavy-weight athletes and 17 American football and rugby non-heavy-weight athletes [27]. Anthropometric data (body mass, height, body mass index (BMI), waist circumference), body composition (body fat percentage, total fat mass, fat-free mass, visceral and subcutaneous fat), liver function (aspartate aminotransferase, alanine aminotransferase, γ-glutamyl transpeptidase), blood lipids (total cholesterol, high-density lipoprotein cholesterol, triglycerides, low-density lipoprotein cholesterol), uric acid metabolism (uric acid concentration), and glucose metabolism (plasma glucose, serum insulin, insulin resistance) were measured [27]. In comparison, Judo athletes showed a significantly higher body mass (122.7 ± 13.1 kg vs. 99.0 ± 8.1 kg), body fat percentage (27.5% ± 5.2% vs. 19.4% ± 4.7%), and higher proportions of visceral fat (118 ± 35 cm^2^ vs. 67 ± 24 cm^2^) than the heavy-weight American football or rugby athletes (all *p* < 0.01) [27]. Nevertheless, the cardiometabolic risk, evaluated through a biochemical analysis of blood, was similar for the heavy-weight Judo and American football and rugby athletes [27]. The only exception was uric acid, which was significantly higher (*p* = 0.026) in the Judo group [27]. Both heavy-weight groups showed an increased cardiometabolic risk when compared to the non-heavy-weight group [27]. The values of alanine aminotransferase (*p* < 0.001), γ-glutamyl transpeptidase (*p* < 0.001), uric acid (*p* = 0.048), fasting plasma glucose (*p* = 0.01), insulin (*p* = 0.01), and homeostasis model assessment insulin resistance (*p* = 0.008) were significantly elevated in the heavy-weight groups. 

### 3.4. Overweight/Obesity

Jung et al. investigated the effects of after-school taekwondo training on cardiovascular disease risk factors in obese male adolescents [28]. Twenty-three participants aged between 13 and 15 years were divided into an intervention and control group. The intervention group trained taekwondo for 60 min, three times a week, for 16 consecutive weeks [28]. The taekwondo program included physical contact, consisting of typical taekwondo movements including punching, kicking, and sparring. Cardiovascular risk factors (blood lipids, blood pressure, arterial stiffness), physical fitness variables (flexibility, muscle strength, muscle endurance, cardiorespiratory fitness), and anthropometric parameters (body mass, height, BMI) were measured before and after the intervention [28]. The intervention group showed a lower cardiovascular risk profile, through improvement of arterial stiffness (pre vs. post right brachial-ankle pulse wave velocity: 995.0 ± 147.6 m/s^−1^ vs. 922.1 ± 97.3 m/s^−1^, *p* < 0.05), total cholesterol (percentage change pre and post taekwondo group vs. control group: −5.8% vs. +8.5%, *p* < 0.05), low density lipoprotein (−8.3% vs. 3.8%, *p* < 0.05), and systolic blood pressure (−6.4% vs. +3.4%, *p* < 0.05), as well as an improvement of their health-connected fitness abilities, namely flexibility, trunk, and lower limb muscle fitness, and cardiorespiratory fitness [28]. Body mass (pre vs. post: 84.6 ± 13.9 kg vs. 80.7 ± 14.9 kg, *p* < 0.01) and BMI (pre vs. post: 28.9 ± 2.4 vs. 27.0 ± 3.0, *p* < 0.05) also decreased significantly in the taekwondo group after the intervention [28].

Another study investigated the effects of twelve weeks of MMA training in premenopausal obese or overweight women [33]. In total, 47 women participated in this study and were randomly assigned to a control or intervention group [33]. The intervention group trained MMA techniques for 60 min, three times a week on non-consecutive days, for a twelve week period [33]. Anthropometric data (body mass, height, BMI), body composition (total fat mass, fat-free mass, total body water, musculoskeletal muscle mass, bone mass), serum biomarkers (serum osteocalcin, c-terminal telopeptide (CTX), insulin-like growth factor 1 (IGF-I), leptin and high-sensitivity C reactive protein (hsCRP), blood lipids (triglyceride, total cholesterol, high-density lipoprotein, low-density lipoprotein), and quality of life were determined [33]. An important finding of this study was that MMA training seemed to improve the quality of life and elevate serum IGF-I levels (average change pre and post martial arts group vs. control group: 9.37 ± 22.78 vs. −5.65 ± 18.38 μg/L, *p* = 0.047) of intervention group members [33]. In terms of body composition, significant changes in the intervention group were observed in fat free mass (−0.004 ± 1.32 vs. 1.33 ± 1.35 kg, *p* = 0.007) and muscle mass (−0.55 ± 2.93 vs. + 1.34 ± 1.34 kg, *p* = 0.022) compared to the control group [33]. 

The study of Roh et al. addressed the effects of regular taekwondo training on body parameters, physical fitness, oxidative stress biomarkers, and myokines in adolescent obese or overweight participants [29]. Fourteen males and six females were randomly assigned to a control or an intervention group [29]. The intervention group participated in 60-min taekwondo training sessions, five times a week for 16 weeks, whereas the control group did not do any specific exercise training, except regular physical education at school [29]. Anthropometric parameters (height, body mass, BMI), physical fitness variables (cardiorespiratory endurance, muscular strength, flexibility, power, balance), levels of serum oxidative stress markers (plasma malondialdehyde, superoxide dismutase), and myokines (serum interleukin-15, brain-derived neurotrophic factor (BDNF), irisin, myostatin) were measured before and after the intervention [29]. The study revealed a significant improvement of physical parameters, in terms of reductions in body mass and BMI, and a significant improvement of muscle strength (leg strength, *p* = 0.034), flexibility (sit-and-reach, *p* = 0.042), and power (surgent vertical jump, *p* = 0.031) in the intervention group as compared to the control group, which did not show any significant improvements [29]. The level of the serum oxidative stress marker superoxide dismutase was significantly higher (pre vs. post: 3.34 ± 0.45 U/mL vs. 3.46 ± 0.48 U/mL, *p* = 0.001) and the level of plasma malondialdehyde was significantly lower (pre vs. post: 5.41 ± 1.40 mmol/mL vs. 4.79 ± 1.21 nmol/mL, *p* < 0.001) in the intervention group after the 16-week intervention [29]. Myokines showed a significant rise in BDNF (pre vs. post: 25.41 ± 5.36 ng/mL vs. 29.52 ± 5.83 ng/mL, *p* = 0.018) and a significant drop in irisin levels (pre vs. post: 145.81 ± 32.18 ng/mL vs. 136.09 ± 28.22 ng/mL, *p* = 0.009) in the intervention group. Serum interleukin-15 (*p* = 0.565) and myostatin (*p* = 0.535) did not differ significantly between the groups [29]. 

Brasil et al. examined the effects of judo training on body composition, cardiac autonomic function, and cardiorespiratory fitness in overweight or obese children [11]. Thirty-five children, aged between 8 and 13 years, took part in this controlled clinical trial [11]. The children were assigned to two groups: one normal weight group with 15 participants (seven females and eight males), and one obese or overweight group with 20 participants (ten females and ten males) [11]. The 60-min judo sessions were conducted twice a week. The overweight or obese group participated in the judo sessions and received physical education at school, whereas the participants in the non-obese or overweight group solely participated in physical education [11]. Anthropometric data (body mass, height, BMI, waist circumference), body composition (total fat mass, fat free mass, body fat percentage, trunk fat), blood pressure, cardiorespiratory fitness (peak oxygen uptake, gas exchange threshold), and cardiac autonomic function (heart rate variability) were measured [11]. Although, the BMI of the children did not change after the intervention, the fat mass (approx. 3%, *p* < 0.05), trunk fat (approx. 3%. *p* < 0.05), and trunk fat/limb fat ratio (approx. 4%, *p* < 0.05) were significantly reduced, whereas the lean mass (approx. 8%, *p* < 0.001) increased in the intervention group [11]. Another finding was a lowered resting heart rate (approx. 3%, *p* < 0.001) after the intervention, while systolic and diastolic blood pressure remained unchanged [11]. Cardiorespiratory fitness improved in the intervention group, with higher relative VO_2peak_ (*p* < 000.1), higher maximum speed in km/h (*p* < 0.05), and an improved oxygen economy at ventilatory threshold 1 during exercise testing (*p* < 0.05). 

The ‘martial fitness’ study of Tsang et al. implemented a randomized placebo-controlled trial, aiming to investigate the effects of Kung Fu training on the metabolic health, physical fitness, and body composition of obese or overweight adolescents [30,31,32]. The study results were presented in three separate papers, each focusing on one of the investigated aspects, namely metabolic health, physical fitness, or body composition. Twenty adolescents, twelve girls and eight boys, aged between 6 and 12 years, were recruited for the study and assigned to an intervention group or a placebo group. Twelve children (five boys, seven girls) were assigned to the intervention group, which had the possibility to practice Kung Fu for 60 min, three times a week, for six months in total. The remaining eight adolescents (three boys, five girls) were assigned to the placebo group, which practiced Tai Chi for 60 min, up to three times a week, for six months overall. The Kung Fu training included non-contact, as well as physical contact, elements, whereas the Tai Chi classes consisted solely of non-contact elements. Anthropometric data (body mass, height, BMI, waist circumference), body composition (total and regional fat, total and regional bone mineral density, lean body mass), blood glucose (fasting glucose, HbA_1c_), blood lipids (high density lipoprotein cholesterol, low density lipoprotein cholesterol, TG, total cholesterol), insulin values (insulin levels, CRP), physical fitness variables (cardiovascular fitness, muscle fitness, maximal muscle strength, peak muscle power, muscle endurance), and habitual physical activity and dietary intake, which were both requested through questionnaires or voice recording. These questionnaires were obtained before and after the intervention [30,31,32]. 

Both programs had some positive metabolic outcomes, even if the Kung Fu program did not result in any superior metabolic changes as compared to the Tai Chi group [31]. Inflammation, shown through levels of CRP (pre vs. post: Kung Fu 3.9 vs. 3.0, Tai Chi 5.9 vs. 4.4, mg/L both *p* = 0.03), fasting insulin, insulin sensitivity, and insulin resistance remained the same in both groups before and after the intervention; only HbA_1c_ (pre vs. post: Kung Fu 5.41 ± 0.32% vs. 5.31 ± 0.35%, *p* = 0.09; Tai Chi 5.43 ± 0.28% vs. 5.38 ± 0.13%, *p* = 0.09) showed a slight, but clinically irrelevant, improvement [31]. The blood lipids also did not change in either group [31]. 

In terms of physical fitness, participants of the Kung Fu group showed greater improvements of upper muscle velocity (*p* = 0.03) and submaximal cardiovascular fitness (*p* = 0.03) in comparison to the Tai Chi group participants [32]. Muscle strength (pre vs. post Kung Fu: lower body 567.0 N vs. 576.0 N, *p* = 0.04; upper body 164.5 ± 26.4 N vs. 178.4 ± 32.1 N, *p* = 0.02; Tai Chi: lower body 464.5 N vs. 534.0 N, upper body 141.8 ± 25.2 N vs. 156.5 ± 28.5 N) and upper-body endurance (pre vs. post: Kung Fu: 5.0 ± 2.5 reps vs. 7.4 ± 3.0 reps; Tai Chi: 5.9 ± 1.9 reps vs. 7.5 ± 2.3 reps, both *p* = 0.02) significantly improved in both groups, whereas peak cardiovascular fitness and muscular power remained unchanged in both groups. There were no significant group differences in muscle strength (lower body: *p* = 0.18, upper body: *p* = 0.93), upper body endurance (*p* = 0.68), peak cardiovascular fitness (*p* = 0.43), and muscular power (lower body: *p* = 0.95, upper body *p* = 0.32) [32].

Both interventions similarly affected the body composition [30]. Due to natural growth, height (*p* < 0.0001), body mass (*p* = 0.0003), total and regional bone mineral density (*p* < 0.0001, *p* < 0.001), and lean mass (*p* < 0.0001) increased significantly in both groups, without significant differences between the groups [30]. 

## 4. Discussion

Our systematic review indicated that regular combat sports have positive effects on health parameters in individuals with metabolic diseases; these studies included results from participants with T1DM, MetS, overweight, or obesity. Our results showed that combat sports are safe and feasible in those with metabolic diseases, since no adverse events related to this specific type of exercise training, particularly in T1DM, were recorded. However, an exceptional finding is that not a single study investigated the effects of combat sports on T2DM. 

Combat sports may be difficult to handle for individuals with T1DM, since the type of exercise is comparable to high-intensity interval exercise, with extended high-intensity bouts and short periods of rest. In the included studies conducting MMA and kickboxing with high physical contact, interstitial glucose monitoring may have been dangerous. Both studies did not state any details about the devices used, if any [9,25]. The study results suggest that participants relied on BG measurements, since the number of daily measurements almost doubled during the study period. These increases in the number of measurements appear to have positively influenced glycaemic variability and HbA_1c_. Nevertheless, the number of weekly symptomatic hypoglycaemic episodes was reduced by 50% from 9.28 ± 2.0 to 4.52 ± 1.6 episodes per week; however, this value is still well above the recommendations from the Juvenile Diabetes Research Foundation (2 symptomatic hypoglycaemic episodes per week) [34]. Although, these studies provide a first insight into the favorable combination of combat sports and T1DM, the findings cannot be accepted as reliable evidence, due to insufficient power. This can be attributed to several limitations in the study designs, e.g., small sample sizes, unspecific exercise regimens without interstitial glucose monitoring, or insufficient statistical processing [25]. Notwithstanding, it is worth underlining that glycaemic control tends to improve with time, probably due to gaining more experience in individual T1DM management while practicing combat sports.

Moreover, the studies investigating MetS were cross-sectional studies, without a detailed intervention period. In the studies from Guo et al. and Murata et al. athletes from professional combat sports were compared with those from other types of sports [26,27]. Due to the weight class system, the prevalence of MetS was much higher in the open weight classes than in the groups that had to cut weight for their individual weight class. In open weight classes that involve grappling and throws, such as Judo and wrestling, body mass has, next to technique and experience, a remarkable influence on performance in competition. Hence, the prevalence of MetS in the upper weight classes is comprehensible. Health-wise, no positive effect, but also no detrimental effect, of combat sports compared to other sports was shown [26]. This finding is supported by Murata et al., who compared Judo athletes, heavy-weight American football and rugby players, indicating the same cardiometabolic risk for all groups, without additional disadvantages for those actively participating in combat sports [27]. This may indicate that even though Judo athletes had a higher body fat percentage and visceral fat, they are not at an increased cardiometabolic risk compared to other comparable groups.

Jung et al. indicated that Taekwondo training seems to be suitable for improvement of cardiovascular disease risk and health-related fitness in their study cohort [28]. It also showed that the regular physical education in schools, as received by the control group, did not have any positive effects on the prevention of cardiovascular disease in the adolescents [28]. A limitation of this study, however, was the small sample size. Additionally, since there were no female participants involved, the outcomes may not be transferable to the other sex.

In post-menopausal obese and overweight women, MMA training may be an appropriate tool for weight management, since the intervention group maintained their body mass, whereas the control group gained weight in the twelve week intervention period [33]. The significance of this study is also limited due to a small sample size and a relatively short intervention time [33]. 

In adolescent cohorts, as shown by Roh et al. and Brasil et al. [11,29], Taekwondo and Judo achieved higher body mass reductions and improvements in cardiovascular fitness compared to the control group. In addition, myokine secretion [29] and cardiac autonomic function improved compared to the control groups [11]. This differs from the findings by Tsang et al. comparing Kung Fu and unstructured Tai Chi lessons [30,31,32]. They favoured Kung Fu in terms of reduced inflammation, and improved physical fitness and body composition. However, insulin and blood lipids remained unchanged, without an advantage over Tai Chi. The informative value of these findings is compromised, since the average attendance for the three exercise sessions per week was 50% in the Kung Fu group and about 40% in the Tai Chi group. Another limitation for the outcomes on body composition were unintended changes in the diet of the children and an inadequate stimulation by intervention, which was not sufficient to achieve changes in body composition [30]. The stimulus-load was too weak in terms of volume (study compliance) and intensity, since the classes were of intermittent character, focusing on teaching the martial art, wherefore the instructor often stopped the classes and explained or demonstrated certain techniques [30].

This systematic review included studies with various cohorts, mostly from unexperienced adolescents, but also post-menopausal women and elite-athletes with decades of experience in combat sports. However, no adverse events, e.g., serious hypoglycaemic events and no cases of deteriorated glycaemic control directly associated with combat sports were recorded. This means that, independent of skill level, combat sports are feasible and may improve parameters of health, as shown in Table 1 and Table 2. Although some combat sports showed higher levels of body fat compared to other athletes, their risk of deteriorating cardiometabolic health was not increased [26]. The included studies in our systematic review also have the potential for further discussion, due to the nature of the examined types of exercise. A subgroup analysis would have been helpful, to further determine the impact of the exercise interventions; yet, because of the diversity of the cohorts, the types of combat sports and vast heterogeneity in the study design, this is not possible without major bias. Future studies investigating this area should consider that unlike other aerobic-anaerobic endurance exercises, combat sports require a high number of skills to adequately conduct the exercise. Most of the studies included unexperienced individuals, who received lessons on how to conduct combat sports and, therefore, the actual exercise duration was shorter than proclaimed in the studies. Consequently, it may be considered that these results, besides those in T1DM [9,25], did not reach the full potential of combat sports for improving the participants’ physiology. Combat sports offer several different methods of exercise training; hence, in future studies the time, exercise, and intensity of, e.g., sparring rounds should be investigated in more detail. In this context, future studies should consider teaching participants this kind of exercise prior to starting an intervention involving combat sports to increase the impact on the physiology and to further strengthen the evidence. This would also apply to future studies involving individuals with T2DM, especially as no study has been conducted so far with T2DM and combat sports. Future studies aiming to investigate this topic should formulate a structured combat sport exercise plan to make the results reproducible and comprehensible for individuals with metabolic diseases, researchers, and general practitioners.

## 5. Conclusions

This systematic review showed that combat sports are safe and feasible in any metabolic disease besides T2DM, due to the lack of conducted studies. Health benefits from performing combat sports include inter alia improved glycaemia, cardiometabolic health, physical fitness, and body composition. Future studies should investigate the effects of combat sports on T2DM in general, glycaemia pre- and post-exercise in T1DM, and structured combat sports regimens in adults who are overweight/obese, with and without MetS.

## Figures and Tables

**Figure 1 jcm-11-01070-f001:**
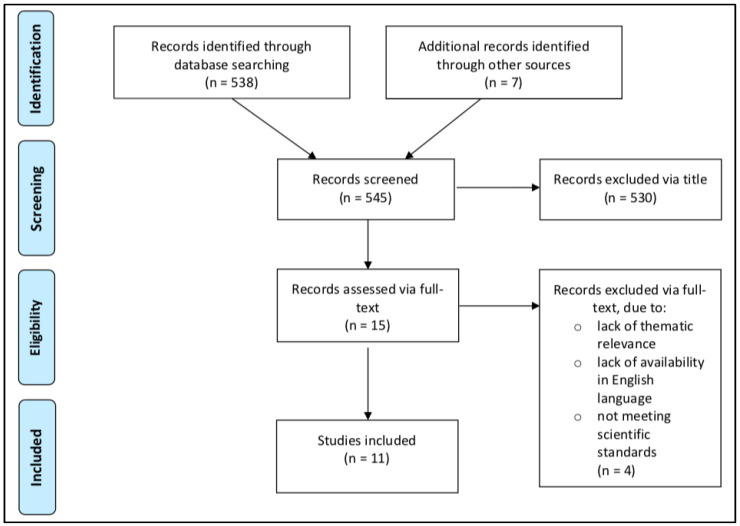
Prisma flow diagram [22].

**Table 1 jcm-11-01070-t001:** Study characteristics, clinically diagnosed individuals with type 1 diabetes and combat sports.

	Year		Disease	*n*	Age (Years)	Combat Sports	Outcome
[9]	2015	2 year follow up Retrospective controlled	Type 1 Diabetes	5	26.0 ± 6.3	MMA/Kickboxing	Reduction in HbA_1c_ by 0.7%, decreased glycaemic variability (84 ± 6 vs. 63 ± 5 mg/dL) and decreased weekly symptomatic hypoglycaemic events (9.0 ± 2.3 vs. 4.5 ± 1.6) after study period
[25]	2017	3 year follow up Retrospective controlled	Type 1 Diabetes	3	28.0 ± 6.5	MMA	Reduction in HbA_1c_, increased daily BG measurements 4.3 ± 1.4 vs. 11.9 ± 0.8

**Table 2 jcm-11-01070-t002:** Study characteristics of clinically diagnosed obesity and metabolic syndrome and combat sports.

	Year		Disease	*n*	Age (Years)	Combat Sports	Outcome
[26]	2013	Cross-sectional study	Metabolic Syndrome	261	21.4 ± 4.9	Judo/Wrestling	Prevalence of metabolic syndrome was significantly higher among highest weight class athletes, compared to lower weight class athletes (89% vs. 18% for males, 47% vs. 0% for females). No advantage of combat sports compared to other sports was shown.
[27]	2016	Cross-sectional study	Metabolic Syndrome	58	20.4 ± 1.1	Judo	Heavy-weight Judo athletes had significantly higher body weight, body fat percentage, and visceral fat (all *p* < 0.01) compared to heavy-weight American football and rugby athletes. Both groups had the same cardiometabolic risk
[28]	2016	Controlled study	Overweight/Obese	23	14.0 ± 0.9	Taekwondo	Cardiovascular risk factors and physical fitness could be improved by the Taekwondo intervention
[33]	2013	Randomized Controlled Trial	Overweight/Obese	47	40.4 ± 6.3	MMA	MMA training improved quality of life and IGF-1
[29]	2020	Randomized Controlled Trial	Overweight/Obese	20	12.6 ± 0.5	Taekwondo	Taekwondo achieved greater weight reduction, improvements of some physical fitness variables and moderation of oxidative stress and myokine secretion compared to control group
[11]	2020	Non-Randomized Controlled Trial	Overweight/Obese	35	11.1 ± 1.1	Judo	Judo exercise improved body composition and cardiorespiratory fitness, and showed a trend towards improved cardiac autonomic function
[30,31,32]	2009, 2010	Randomized Controlled Trial	Overweight/Obese	20	13.1 ± 2.1	Kung Fu	Kung Fu training reduced inflammation. Insulin and blood lipids remained unchanged. Physical fitness variables, except peak cardiovascular fitness, were improved. Body composition improved as well.

## Data Availability

Data may be obtained from the corresponding author upon reasonable upon request.

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
