# Peer review of "Physiological Responses to Combat Sports in Metabolic Diseases: A Systematic Review"

_jcm, 2022, doi:10.3390/jcm11041070_

Round 1
Reviewer 1 Report
It is thought that an appropriate analysis of the detailed process related to the positive effects of combat sports has been made.
However, please describe the clear rules of combat sports in the research method.
Discussion: It is necessary to consider the detailed effects of each type of combat sport (intensity, exercise time, exercise type, etc.).
Author Response
Dear Reviewer 1,
It is thought that an appropriate analysis of the detailed process related to the positive effects of combat sports has been made.
Thank you very much for your comments and your kind words about our manuscript.
However, please describe the clear rules of combat sports in the research method.
Thank you very much for your comment. We acknowledged the research by Krabben et al. regarding the definition of combat sports. We updated this section now so it’s clearer regarding the styles and rules of this diverse type of exercise.
Line 116 – Line 125
For this review we focused on the definitions of combat sports according to Krabben et al.. Combat sports covers disciplines, where two opponents fight in a regulated combat, trying to win over their adversary through usage of strikes, kicks, or grappling in different configurations (23). The main difference between martial arts and combat sports is the necessarily included physical contact in combat sports which is not a mandatory requirement to martial arts (24). Principal items of martial arts are sets of formal exercises and expressing movements, which not always include physical contact (24). Consequently for this review, combat sports were defined as any kind of martial arts involving physical contact involving striking, kicking, wrestling or grappling. Studies including martial art without physical contact or types of kata against imaginary opponents or stylistic sequences of presenting martial arts techniques were excluded.
Discussion: It is necessary to consider the detailed effects of each type of combat sport (intensity, exercise time, exercise type, etc.).
Thank you very much for your comment. We agree with the reviewer and added/amended this in the discussion. We hope this is acceptable for the reviewer.
Line 427 – Line 438
Future studies investigating should consider that unlike other aerobic-anaerobic endurance exercises, combat sports require a high number of skills to adequately conduct the exercise. Most of the studies included unexperienced individuals who received lessons on how to conduct combat sports and therefore, the actual exercise duration was shorter than proclaimed in the studies. Consequently, it may be considered that these results, besides those in T1DM (9,25), did not reach the full potential of combat sports in improving the participants’ physiology. Combat sports offer several different methods of exercise training, hence in future studies the time, exercise and intensity of e.g. sparring rounds should be investigated in more detail. In this context, future studies should consider teaching participants this kind of exercise prior to starting an intervention involving combat sports to increase the impact on the physiology and to further strengthen the evidence
Reviewer 2 Report
In my opinion, the article in current form is suitable for publication.
Author Response
Thank you very much for your time to read through our manuscript again. Thank you for your kind words.
Reviewer 3 Report
In my opinion, the revised manuscript is worthy to be published because the authors responded to the requested revisions.
Moreover, this study demonstrates that also combat sports can improve metabolic parameters and can be performed safely by patients with metabolic diseases.
Author Response
Thank you very much for your time to read through our manuscript again. Thank you for your kind words.
This manuscript is a resubmission of an earlier submission. The following is a list of the peer review reports and author responses from that submission.
Round 1
Reviewer 1 Report
The Authors conducted a systematic review on the feasibility, safety and benefits of combat sports in patients with metabolic diseases. They showed that performing combat sports has some health benefits such as improved glycaemia and the reduction of hypoglycaemic episodes in type-1 diabetes mellitus and the amelioration of body composition, the reduction of cardiometabolic risk in overweight/obese patients.
In my opinion, this paper addressed some important points on the topic.
- Combat sports have always been identified as sports with a high risk of injury and with small health benefits. Instead, this review showed that also patients with metabolic diseases, can take advantages of combat sports with the improvement of metabolic parameters and can perform them safely.
- This review showed the limits of the study evaluating combat sports, such as the small sample size and the heterogeneity of studied populations. Moreover, it underlines that there are no studies on the role of combat sports in NAFLD or type-2 diabetes mellitus, the most widespread metabolic diseases. So, this review could be a stimulus to conduct more and well-structured studies on the topic.
Minor revisions:
In general, the paper is well written, but some minor language details should be corrected. For example:
- Page 4 (line 168), page 5 (line 195), page 6 (lines 253 and 276): after the point the numbers should be written in letters
- Page 5, line 190: after p<0.001 a parenthesis is needed
Author Response
Reviewer 1:
“The Authors conducted a systematic review on the feasibility, safety and benefits of combat sports in patients with metabolic diseases. They showed that performing combat sports has some health benefits such as improved glycaemia and the reduction of hypoglycaemic episodes in type-1 diabetes mellitus and the amelioration of body composition, the reduction of cardiometabolic risk in overweight/obese patients.
In my opinion, this paper addressed some important points on the topic.
- Combat sports have always been identified as sports with a high risk of injury and with small health benefits. Instead, this review showed that also patients with metabolic diseases, can take advantages of combat sports with the improvement of metabolic parameters and can perform them safely.
- This review showed the limits of the study evaluating combat sports, such as the small sample size and the heterogeneity of studied populations. Moreover, it underlines that there are no studies on the role of combat sports in NAFLD or type-2 diabetes mellitus, the most widespread metabolic diseases. So, this review could be a stimulus to conduct more and well-structured studies on the topic.”
Thank you for taking your time to evaluate our manuscript. We would also like to thank you for your supportive comments.
„Minor revisions:
In general, the paper is well written, but some minor language details should be corrected. For example:
- Page 4 (line 168), page 5 (line 195), page 6 (lines 253 and 276): after the point the numbers should be written in letters
Thank you for highlighting this. This has been amended.
- Page 5, line 190: after p<0.001 a parenthesis is needed”
Thank you for pointing this out. This has been amended.
Reviewer 2 Report
This paper has interesting and important information about the physiological responses to combat sports in metabolic diseases: type 1 or 2 diabetes mellitus, metabolic syndrome, overweight and obesity. This systematic review was well written and clearly shows that martial arts are safe and feasible in the metabolic diseases analyzed. The health benefits of practicing martial arts include many parameters, such as improving glycemia, blood presure and body composition, among others.
There are some issues in References. Make certain that all of the references are correct (e.g. item 19 - full title of the journal, no volume; item 27 - no volume and number of pages).
Author Response
Reviewer 2
„This paper has interesting and important information about the physiological responses to combat sports in metabolic diseases: type 1 or 2 diabetes mellitus, metabolic syndrome, overweight and obesity. This systematic review was well written and clearly shows that martial arts are safe and feasible in the metabolic diseases analyzed. The health benefits of practicing martial arts include many parameters, such as improving glycemia, blood presure and body composition, among others.
There are some issues in References. Make certain that all of the references are correct (e.g. item 19 - full title of the journal, no volume; item 27 - no volume and number of pages).”
Thank you for your comments. We have reviewed the references and amended missing features in the reference list. Reference 19 and reference 27 have been corrected. We have furthermore added additional literature to the reference list to give more information within the introduction.
Reviewer 3 Report
Please provide a more clear rationale for analyzing the relationship between Combats sports and metabolic syndrome-related indicators in the need for research.
In the composition of the detailed table of contents of the study, it is thought that a clearer rationale for classifying type 1 diabetes, metabolic disease, and obesity index should be presented.
Fundamental improvement of research design in this area is required. For example, it is recommended that the study design be redesigned by dividing it into detailed analysis of metabolic disease-related risk indicators.
Author Response
Reviewer 3
Thank you for taking your time and meticulous work on our manuscript. Please find below a point-to-point response letter to your specific queries. Overall, we hope that our revisions are in agreement with the reviewers’ suggestions.
„Please provide a more clear rationale for analyzing the relationship between Combats sports and metabolic syndrome-related indicators in the need for research.”
Thank you for your comment. We have added a clearer rationale to the introduction. We hope this agrees with the reviewers’ suggestion.
“The prevalence of metabolic diseases, such as the metabolic syndrome (MetS), type 1 diabetes mellitus (T1DM), type 2 diabetes mellitus (T2DM) and obesity have been rising over the past decades (1–3). These diseases have a major impact on the physiology and favor the development of comorbidities that may further deteriorate quality of life by reducing physical capacity, deteriorating bone quality and increasing the risk of cardiovascular disease (4–6).
For adults aged between 18 - 64 years, the World Health Organization (WHO) recommends at least 150 minutes of moderately-intense aerobic physical activity or 75 minutes of vigorously-intense aerobic physical activity or a suitable combination of both every week (7). Low to moderate physical activity is usually considered as any kind of recreational activity of aerobic nature such as walking, hiking or cycling.
Besides, the interest in conducting combat sports, defined as physical contact between two opponents in a martial art manner, has increased rapidly during the last decades. Many people enjoy doing combat sports for recreational purposes. About 2% of the physically active European population practiced martial arts in 1999 making martial arts the 10th most popular sport next to hill walking/climbing (8). Also, children and adolescents in particular practice combat sports, such as taekwondo or judo (9–11). This type of sport is an interval-based physical activity, which is conducted in groups or with a partner. The intensity during the training sessions may vary, yet it is mostly of moderately intense aerobic nature with short bouts of vigorous anaerobic intensity. At the same time, the positive effects of high-intensity interval training on individuals with metabolic diseases improving glycaemia, body composition and fitness, have already been investigated thoroughly (12–14).
The emerging prevalence of metabolic diseases, and an increasing interest of many people in combat sports lead to the question whether combat sports, which cover several types of differently intense training methods, are safe and applicable for people affected by a metabolic disease. Previous studies have shown that combat sports increase physical capacity (15) and improve bone mineral density (16,17).
One might wrongly assume that combat sports have a higher risk of injury such as sprains, fractures or concussions compared to team sports such as football. However, according to Arriaza et al. and Emery et al., 0.67 injuries occur every 1.000 minutes during combat sports in comparison to 5.59 injuries every 1.000 minutes during amateur football (18,19). A further advantage of combat sports compared to other sports is the reduced level of aggressiveness of individuals actively practicing combat sports to diffuse emotions, relieving tension to improve self-esteem and quality of life (20).
According to Hamasaki et al., no positive effects on metabolic diseases by Tai Chi practice could be proven. In contrast other martial arts performed as Kata, e.g. as Kung Fu or Karate demonstrated some positive effects on body composition, glycaemic control, and arterial stiffness (21). Due to the specific nature of combat sports (whole body movements at various intermittently conducted intensities) it is of interest if individuals with metabolic disease may benefit from regularly conducting this type of exercise by improving parameters of metabolic disease and/or reducing the development of comorbidities. Consequently, the aim of this systematic review was to investigate how individuals with metabolic disease respond to combat sports and if those are feasible, safe and applicable in daily life. Unlike numerous other reviews investigating the interaction of martial arts and metabolic disease with contrary outcomes, the focus of this systematic review will lie upon combat sports.”
In the composition of the detailed table of contents of the study, it is thought that a clearer rationale for classifying type 1 diabetes, metabolic disease, and obesity index should be presented.
Thank you for your comment. We entirely agree with you. However, due to the style of article we are prohibited to either add or remove information given in the cited paper as the approach of a systematic review is to summarize the given data within the studies and to put them into context with the results of the other included studies. We have noticed that the presentation of the results in several studies could be improved. Hence, we have mentioned this in the limitations section in the discussion. In particular the different outcome criteria and specifications of disease demand more detailed information in future studies.
Yet, after reviewing all studies carefully again. We can confirm that all included populations were clinically pre-screened according to national guidelines from the investigators. This has been added to the table legends.
Fundamental improvement of research design in this area is required. For example, it is recommended that the study design be redesigned by dividing it into detailed analysis of metabolic disease-related risk indicators.
Thank you for your comment. We agree with you that generally more research should be conducted in this field of research. In regards to our research design we partly agree with the reviewer that detailed analysis’ e.g. subgroup analysis would be helpful. This would demand additional statistical calculations that are for several reasons not applicable. Due to the diversity of the types of combat sports, types of populations and parameters that were investigated a subgroup analysis would not be possible without major bias. This would also need more studies with more balanced sample sizes for each cohort and/or each specific outcome variable, which is due to the lack of studies not possible. For a detailed analysis a single outcome parameter for each population should have been defined prospectively. Yet, since this is the first review, independent of systematic or narrative, on this topic it was not possible to set outcome parameters for the single populations due to the diversity of the study designs prospectively. Furthermore, the aim of this article was to perform a systematic review of the literature not a meta-analysis on specific risk factors influencing disease via combat sports. Future systematic reviews with meta-analysis should focus on a single population with a distinct health related parameter associated with metabolic disease and a single combat sport.
We hope this is acceptable for the reviewer and we apologize if this was misleading within our manuscript.
Subgroup analysis would have been helpful to further determine the impact of the exercise interventions, yet because of the diversity of the cohorts, the types of combat sports and vast heterogeneity in the study design this is not possible without major bias. Future studies investigating should consider that unlike other aerobic-anaerobic endurance exercises, combat sports require a high number of skills to adequately conduct the exercise.